# Estimating the incidence of dengue in international air travelers from non-endemic countries between 2010–2019

**Matt D. T. Hitchings**[1,2], **Yi Xu**[1], **Bernardo García-Carreras**[2,3], **Adriana Gallagher**[2,3], **Justin J. O'Hagan**[4]*, **Derek A. T. Cummings**[2,3]

**1** Department of Biostatistics, College of Public Health & Health Professions, University of Florida, Gainesville, Florida, United States of America, **2** Emerging Pathogens Institute, University of Florida, Gainesville, Florida, United States of America, **3** Department of Biology, University of Florida, Gainesville, Florida, United States of America, **4** Merck & Co., Inc., Rahway, New Jersey, United States of America

* justin.ohagan@merck.com

## Abstract

### Background

There have been increasing numbers of travel-associated dengue cases reported but the true burden is unclear. Existing surveillance in non-endemic countries captures only a fraction of symptomatic cases in returning travelers due to underreporting. Therefore, we used mathematical modeling approaches to account for underreporting and estimate the number of dengue cases occurring during international travel.

### Methodology/principal findings

We obtained data on numbers of international air passengers from 43 non-endemic "origin" countries, risks of infection while in 119 dengue-endemic locations ("destinations"), and average durations of stay. We estimated travel-associated infections by multiplying the time spent by travelers in endemic countries by the risk of dengue infection and used data on reported cases to infer the fraction of cases that are included in surveillance systems. Our model estimated there were an average of 64,623 (95% CI: 25,068–138,283) symptomatic dengue cases ("cases") and 303,870 (95% CI: 292,841–315,240) total dengue infections (i.e., including symptomatic and asymptomatic infections) across 43 origin countries annually between 2010–19. The USA had the highest number of estimated cases followed by China. Among 34 origin countries that reported dengue cases, the fraction of cases reported varied widely (median 24.9%, range 2.5%-100%). The destination countries where most cases were infected were India, followed by Thailand.

**Data availability statement:** The OAG flight data are proprietary and can be purchased from www.oag.com. ArboNET is the United States Centers for Disease Control and Prevention national arboviral surveillance system, which includes information about confirmed and probable dengue cases identified by healthcare providers and laboratories reported to state and territorial health departments. To maintain patient confidentiality and ensure users understand the limitations of the data, the dataset used here is only available with a data use agreement from the CDC Division of Vector-Borne Diseases. To request the data contact dvdib2@cdc.gov. Dengue surveillance data in ArboNET surveillance is publicly available in aggregated format (https://www.cdc.gov/dengue/data-research/facts-stats/historic-data.html). Data from ECDC's TESSy can be requested (https://ecdc.europa.eu/en/publications-data/european-surveillance-system-tessy). Other data obtained from publicly available sources are available in the Supplementary Materials.

**Funding:** This study was funded by Merck Sharp & Dohme LLC, a subsidiary of Merck & Co., Inc., Rahway, NJ, USA. The funder provided support through salaries/stock options to JO; fees were paid to University of Florida(which employed MDTH, YX, BGC, and DATC). The funder had input into the study design, data collection, analysis, manuscript preparation, and decision to publish.

**Competing interests:** I have read the journal's policy and the authors of this manuscript have the following competing interests: JOH is an employee of Merck Sharp & Dohme LLC, a subsidiary of Merck & Co., Inc., Rahway, NJ, USA and may hold stock or stock options in Merck & Co., Inc., Rahway, NJ, USA. DATC reports a contract from Pfizer (to the University of Florida) for research unrelated to this article. The other authors have declared that no competing interests exist.

## Conclusions/significance

We estimated a substantial burden of dengue among international air travelers from 43 non-endemic origin countries. The fraction of cases reported varied widely across origin countries and was also influenced by the specific origin-destination country pair examined.

## Author summary

Thousands of dengue cases are reported in non-endemic countries each year due to returning travelers having been infected in tropical and subtropical regions. However, the true number of dengue illnesses among travelers is unknown because not all cases seek medical care and an unknown proportion of those who do are not diagnosed or reported to health authorities. Consequently, travelers and their medical providers do not have accurate information on the risk of acquiring dengue during trips to inform preventative behaviors. To address this gap, we estimated the incidence of dengue among travelers from 43 non-endemic countries between 2010–19. We found there were an average of 64,623 symptomatic dengue cases across 43 origin countries annually between 2010–19. The USA had the highest number of estimated cases (16,551) followed by China (7,298). Across considered countries, a median of 24.9% of cases were reported to health authorities. Our findings can be used by travelers and their healthcare providers during pre-travel visits to inform about the risks of dengue.

## Introduction

Among arthropod-borne viruses, dengue virus causes the highest burden globally with half the world's population living in areas that put them at risk of infection and an average of 50–60 million symptomatic cases occurring annually [1–3]. Dengue is also commonly among the top four most diagnosed pathogens among international travelers after returning home [4–6].

Travel medicine experts emphasize the need for updated assessments of the risks of travel illnesses to guide prioritization of preventive strategies [6]. Similarly, the most common reason why international travelers do not seek or adhere to pre-travel medical advice is a low perceived risk of illness [7–9]. Consequently, an understanding of the incidence of dengue in travelers visiting endemic regions can inform public health decisions, including tailoring messaging to travelers and clinicians about risks and recommendations about personal preventative measures [10], as well as contributing to societal understanding of the potential negative consequences of international travel beyond the impacts on carbon emissions [11–13]. Furthermore, such information is particularly important for non-endemic countries with suitable vectors (e.g., USA, southern Europe, Japan, Hong Kong) as returning travelers with active infection can spark local outbreaks [14–19].

Sentinel surveillance systems report notified cases of dengue and are useful for detecting changes in incidence among travelers over time [20–22]. However, surveillance systems cannot provide robust estimates of the absolute risks of dengue infection nor illness because up to 80% of dengue infections are asymptomatic [23] and most symptomatic cases (hereafter termed "cases") are not reported even in countries with robust surveillance (e.g., due to lack of care-seeking, misdiagnosis, lack of adherence to reporting requirements) [24,25].

Seroepidemiological studies, in which paired blood samples from travelers are used to infer exposure to a pathogen, can provide estimates of infection risk [26–28]. However, these studies suffer from bias due to false positives given that the true incidence is likely low [29]. Also, they cannot feasibly provide risk estimates for all traveler origin and destination country pairs of interest, nor be conducted regularly due to being resource-intensive given the sample sizes required.

Several previous studies have used mathematical models to quantify the risk of dengue importation into dengue non-endemic countries from air passengers [30–41]. These models vary in complexity, but generally rely on estimates of dengue incidence in source countries and traveler volumes to estimate importation risk. Most of these studies focused on single countries [31,33–35,38] or on European countries [30,37,40], and on a subset of endemic destination countries from which most travel-associated cases are reported [31,32,34,35,38–40]. Several studies used reported case data in travelers to estimate reporting rates or validate models [30,31,35,36,38].

In this paper, we aim to extend existing models by jointly estimating potential travel-associated dengue cases across 43 non-endemic ("origin") countries and 119 endemic ("destination") countries between 2010–19. By including this large number of countries, we assessed consistency and patterns in the fractions of potential traveler dengue cases reported across non-endemic countries and explored how these reporting fractions varied based on travelers' origins and destinations.

## Methods

### Overview

We conducted three principal analyses for the period 2010–19 to estimate: 1) total potential dengue infections (i.e., cases and asymptomatic infections) and symptomatic infection or cases among travelers, 2) case reporting fractions by origin country (i.e., percent of estimated dengue cases among travelers from an origin country that were reported by that origin country's surveillance system), and 3) case reporting fractions by origin and destination country. We did not conduct analyses for 2020 or later due to the large and temporary impacts of the SARS-CoV-2 pandemic on international travel volumes and dengue incidences in endemic countries [42–44] and because of less availability of data from 2020 onwards.

We provide a high-level summary of the analyses below with technical details available in the Supplementary Material.

**Analysis 1 (Estimating total infections and cases).** We estimated the average annual numbers of potential total dengue infections and cases among international travelers between 2010–19 for a set of 43 origin countries (i.e., non-endemic countries where travel-associated cases were reported). To accomplish this, we multiplied the following three quantities: numbers of airline travelers from each origin country to each dengue-endemic "destination" country, their average durations of stay at the destinations, and the annual hazards of dengue infection experienced in the destination countries. We informed our estimates of time-varying hazard in endemic countries using reported case data, but as the case symptomatic proportion may vary across years, annual case numbers may not accurately reflect changes in transmission intensity [45]. To understand the robustness of the country rankings and overall case numbers to this assumption, we performed a separate analysis assuming that dengue infection hazard was constant over 2010–2019 in each destination country.

**Analysis 2 (Estimating case reporting fractions by origin country).** We estimated the proportions of the potential cases estimated in Analysis 1 that were reported in each origin country using the data above and surveillance reports on the total numbers of travel-associated cases recorded in each origin country. We did this for a subset of 34 countries that had publicly available data on the annual number of travel-associated dengue cases for at least one year between 2010 and 2019. We estimated these quantities using negative binomial regression without an intercept and with the log of the

expected number of cases for each origin country as an offset term. The outcome variable was the observed case counts in origin countries that reported travel-associated cases. This framework does not impose a constraint on the proportion of potential cases that were reported, meaning that it can be above one, which is more likely when the true number of infections is small due to random variation.

**Analysis 3 (Determining global patterns in variation of reporting fractions across origin and destination countries).** We analyzed how origin countries' reporting fractions varied by destination country (e.g., clinicians may be more likely to test a febrile returned traveler for dengue if their destination was one that is widely known to be dengue endemic). To do this, we examined 22 out of 43 origin countries that reported information on the destinations where traveler cases were infected and combined this information with the data employed in the previous two analyses. Briefly, we estimated the number of reported travel-associated dengue cases with known country of acquisition by multiplying the number of infections estimated in Analysis 1 by the fraction of dengue infections among travelers that are reported and have known country of acquisition. In this analysis, we estimated infection (not case) reporting fractions because our primary interest was in exploring the variation across countries. Specifically, by focusing on total infections instead of cases, we improved precision by omitting the variable for the proportion of cases that are symptomatic from calculations and its associated uncertainty.

Finally, we performed a leave-one-out validation to understand patterns in which origin-destination country pairs showed poor goodness of fit or were otherwise inconsistent with other country pairs. We fitted linear regression lines to the predicted vs. observed case numbers and defined outliers as points with Cook's distance greater than four times the average (Cook's distance measures the influence of removing a single point). In this analysis, points that lay above the regression line indicated that more cases were predicted than observed, meaning that travelers in that country pair were at lower risk than the endemic force of infection (FOI) predicted or that reporting probabilities in travelers from that country pair were lower than average.

## Data sources

Our analyses relied on five data sources described below.

A. Surveillance data from origin countries: We included 43 origin countries and territories in analyses, primarily members of the European Union as well as non-EU high-income countries and territories in Europe, North America, Southeast Asia, and the Middle East. We also included China due to the large volume of air travelers from this country. We sought publicly available dengue surveillance data from 43 origin countries (see S1 Table for a full list of origin countries), including information on the likely country of infection. We obtained data from the European Centre for Disease Prevention and Control (ECDC) [46] on all current members of the European Union that reported dengue data between 2010 and 2019, and data on dengue cases in US travelers from the CDC's ArboNET surveillance reporting system [47]. For other countries, we obtained available data from published papers and reports [48–57] (see S1 Table for a full list of data sources). S2 Table contains all reported cases by origin and destination country, excluding case data from CDC's ArboNET, which are not publicly available. Data from 9 out of 43 (20%) of countries could not be obtained, and for a further 12 countries there were no available data on country of infection. Of the 22 countries with data on the destinations of cases, 5 had data on all years from 2010-2019 (Germany, Japan, New Zealand, Taiwan, and the USA), while the remaining had data covering part of that period.

B. Infection risks in destination countries: To quantify the dengue infection risk in endemic destination countries, we used estimates of their FOI from the Global Dengue Transmission Map (GDTM) [3]. An annual FOI can be interpreted as the proportion of fully susceptible individuals in a population that would be infected over the course of a year. These estimates are based on 382 independent serosurveys or surveillance data sets and use data on climatic suitability to interpolate FOI globally to a 1/6-degree pixel resolution (approximately 20km by 20km), together with an associated

standard deviation. The US CDC lists the dengue endemicity statuses for 148 countries or territories [58], and we applied the following exclusion criteria to create a final list of 119 destinations: countries or territories with no FOI estimate in the GDTM map due to lack of serosurveys or available case data or due to sporadic outbreaks (n=24); and countries included in the list of origin countries due to lack of data on within-country travel (n=5). For this list of 119 destination countries (S3 Table), we calculated the dengue FOI experienced by travelers and associated standard deviation as the resident population-weighted estimate of pixel-specific FOI and standard deviation from GDTM, making the assumption that travelers are more likely to travel to locations with greater local populations. Notably, 103/119 (87%) of destinations were listed in origin countries' dengue surveillance data as locations where travelers were infected. In addition, since the publication of the GDTM, a systematic review [59] has identified further age-stratified serosurveys together with estimated FOI from 30 endemic countries. We examined the consistency of these estimates with our GDTM estimates (S1 Fig in S1 Text) and included data on national FOI estimates from the Vicco et al. review where available.

C. Annual case data in epidemic countries: The GDTM provides average annual hazard estimates, but dengue incidence typically displays year-to-year variation. To quantify this variation we obtained publicly available data on reported dengue incidence in endemic countries from OpenDengue [60]. This data set covers 70/119 endemic countries and territories covered by this analysis, and we excluded the data from Pakistan and Vietnam from this data set due to missing years of data. To estimate annual FOI, we used annual case data in country $j$ and year $t$, $c_j^t$, together with the GDTM average FOI to estimate annual FOI $\lambda_{jt} = \lambda_j * \frac{c_j^t}{\sum_t c_j^t}$. Finally, we used available age-stratified seroprevalence data from Puerto Rico [61], the US Virgin Islands [62], and American Samoa [63] to inform annual hazard of infection in these three territories.

D. Travel volumes: We obtained data on the annual numbers of travelers going from origin to destination countries between 2010-19 from the company OAG (Official Aviation Guide) [64]. These data included the numbers of individuals who took commercial airplane journeys between all pairs of origin and final destination countries (i.e., layovers are not included as origin-destination trips in this database).

E. Average trip durations: We used data from the United Nations World Tourism Organization (UNWTO) World Tourism Barometer [65] to obtain information on international travelers' average lengths of stay in destination countries for each year between 2010–19. This source provided data only on average duration by destination country or by origin country. Consequently, we used duration by destination country and assumed duration was independent of origin country. For the 32 destination countries (28% of all destination countries) that reported any data on duration of stay, 26% of years were missing data. For years with missing data in these 32 countries, we used the nearest year for that country to impute trip duration. For the other 83 countries (72% of destination countries), we used the average across all other destination countries and years to impute trip duration. Data on trip duration from UNWTO used for this study are displayed in S4 Table.

## Results

### Analysis 1: Estimated numbers of air travel-associated dengue cases vary by origin country

Using data on airline travel volumes, trip durations, and destination countries' dengue FOIs, we estimated there was an average of 64,623 (95% CI: 25,068–138,283) air travel-associated potential dengue cases annually between 2010–19 across all origin countries (Fig 1, S5 Table). This corresponded to an annual average of 303,870 (95% CI: 292,841–315,240) total potential dengue infections across 43 origin countries.

Of the 43 origin countries, the USA had the highest estimated air travel-associated cases (16,550; 95% CI 6,432–35,560) and total infections (78,947; 95% CI 70,951–87,870), followed by China with 7,298 cases (95% CI 2,800–15,665)

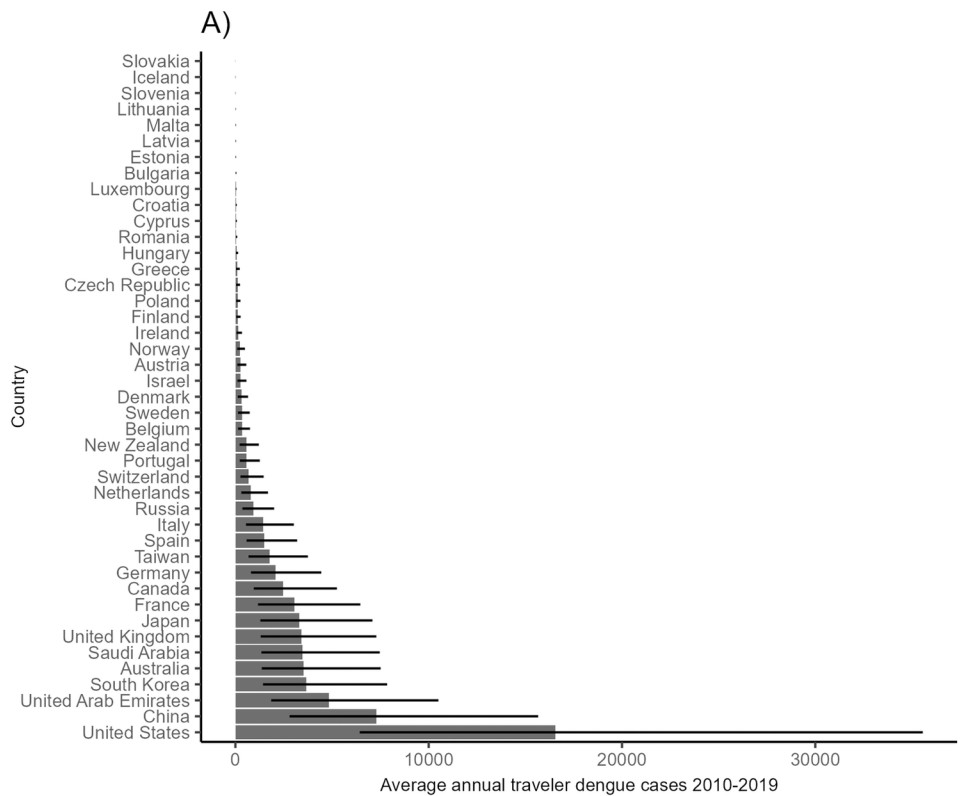

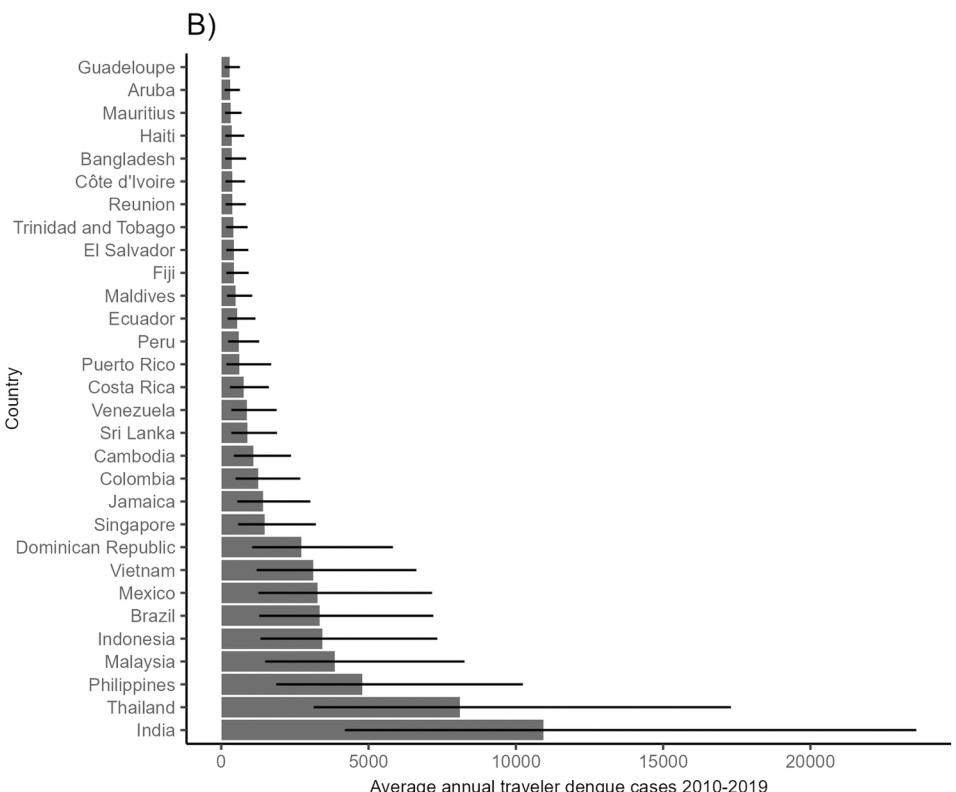

**Fig 1. Estimated annual average number of air travel-associated dengue cases (2010-19) across all 43 origin countries (A). In (B), the 30 destination countries with the highest travel-associated cases are shown. Median cases across 10,000 realizations of endemic force of infection**

**and probability of apparent disease are shown as bars, with 2.5% and 97.5% bootstrap quantiles shown as error bars. (Results used to plot figures provided in S5-S6 Tables).**

and 34,084 total infections (95% CI 30,737–37,424). United Arab Emirates, South Korea, Australia, Saudi Arabia, United Kingdom, Japan, France, Canada, and Germany were the next ranked origin countries with each having between 2,000 and 5,000 estimated annual cases (S5 Table). The destination country that was associated with the most cases was India (10,940; 95% CI 4,194–23,596), followed by Thailand (8,102; 95% CI 3,137–17,302), with Philippines, Malaysia, Indonesia, Brazil, Mexico, Vietnam, and Dominican Republic, and the next ranked destinations with each giving rise to between 2,000 and 5,000 cases annually (S6 Table). A sensitivity analysis in which endemic countries were assumed to undergo constant annual FOI returned similar results for most countries, as travel patterns did not change significantly from 2010-2019 in many of the 32 origin counties (S7 Table). Finally, in a sensitivity analysis restricted to the years 2010–2015, preceding the Zika epidemic which may have affected FOI estimates, reported dengue incidence from endemic countries, as well as traveler reporting patterns, we estimated a total of 46,600 annual cases (95% CI: 18,500–96,750) (S8 Table).

## Analysis 2: Estimated reporting fractions by origin country

Among the 34 countries that reported total numbers of travel-associated dengue cases, there was a strong positive correlation between annual predicted cases and reported cases (Pearson correlation 0.56; S2 Fig). We estimated the average reporting fractions across years for which cases were reported (Figs 2, and S3 in S1 Text, S9 Table). The median reporting fraction across the 34 origin countries was 24.9%.

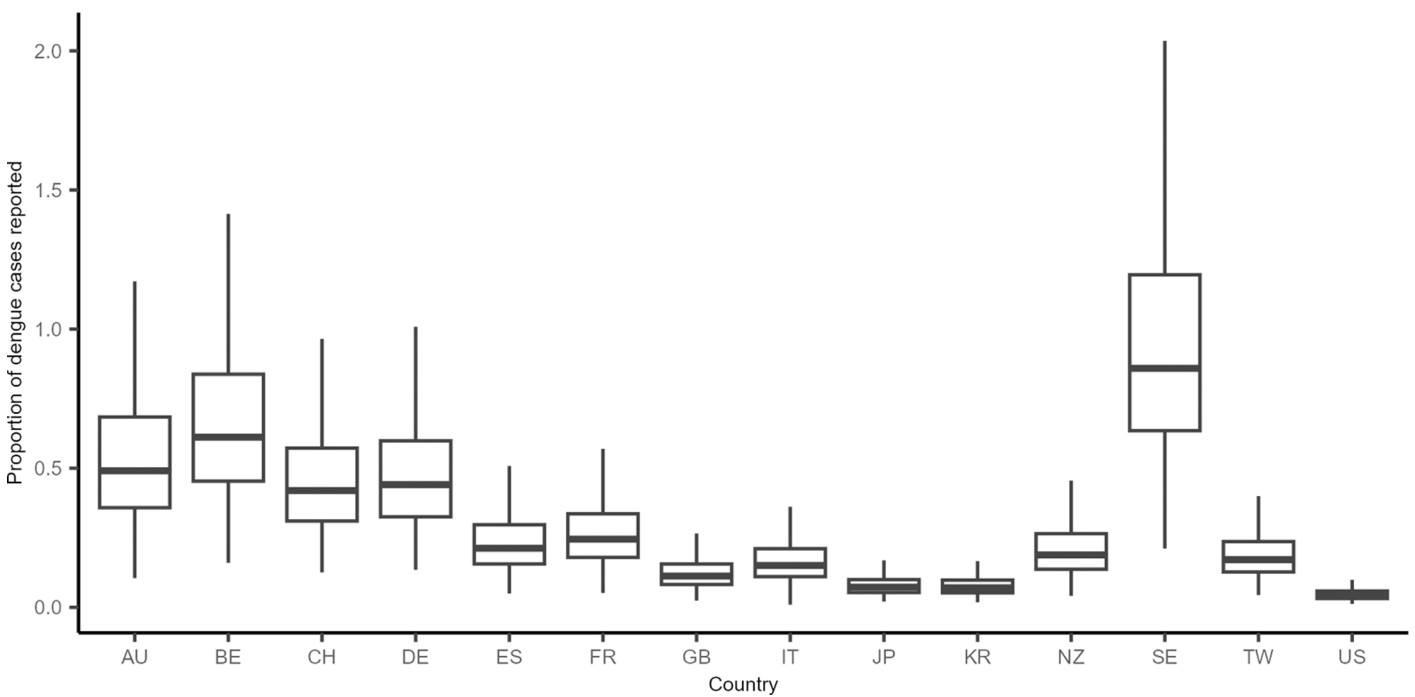

**Fig 2. Estimated fraction of dengue cases reported to national surveillance systems between 2010-19 among 14 origin countries reporting at least 100 dengue cases annually, accounting for uncertainty in endemic force of infection (FOI) and probability of symptoms.** AU: Austria; BE: Belgium; CH: Switzerland; DE: Germany; ES: Spain; FR: France; GB: United Kingdom; IT: Italy; JP: Japan; KR: South Korea; NZ: New Zealand; SE: Sweden; TW: Taiwan; US: United States of America.

Countries with >100 annual reported cases with the highest reporting fractions included Sweden (reporting fraction 85.9%, 95% CI 38.9-231.4), Belgium (61.2%, 95% CI 28.1-158.0), Australia (49.1%, 95% CI 21.9-130.3), and Germany (44.1%, 95% CI 20.4-114.7). Countries with >100 annual reported cases with the lowest reporting fractions included the United Kingdom (11.3%, 95% CI 4.9-29.4), Japan (7.3%, 95% CI 3.3-19.3), South Korea (7.0%, 95% CI 3.1-18.8), and the USA (4.3%, 95% CI 2.0-11.2). However, the confidence intervals were wide in many instances.

### Analysis 3: Infection reporting fractions varied by origin and destination country

There was high variation in infection reporting fractions across country pairs with wide credible intervals associated with many country pairs due to sparse data (S10 Table). When assessing changes in infection reporting fractions driven by origin countries, we identified a set of countries with lower-than-average reporting fractions (Italy, South Korea, Japan, United States, and Spain), while Sweden, Australia, Germany, Taiwan, and New Zealand had higher-than-average reporting fractions. Among destination countries, we found that countries including Singapore, Colombia, Brazil, and Mexico had a lower-than-average fraction of estimated potential cases reported when travelers returned to their origin countries, while Bangladesh, Pakistan, and Sri Lanka were countries for which a higher proportion of estimated potential cases were reported (S11 Table).

### Patterns in reporting by country pairs

To explore patterns in under- and over-predicting of travel-associated cases by destination, we performed a leave-one-out validation for 100 country pairs. Plots of observed versus predicted cases for the left-out pairs are shown by year in Fig 3, with outliers highlighted in red. Country pairs for which the endemic country is a small island nation are over-represented among outliers (62% of outliers compared to 28% of all pairs, p < 0.001 by $\chi^2$ test of independence). In a sensitivity analysis in which we assumed constant FOI in endemic countries, we found a similarly high proportion of island nations among outliers (61%), while we observed many years in which high numbers of cases were predicted to occur among travelers despite zero cases being observed (S4 Fig in S1 Text, along the y-axis for each panel), for example among German travelers to the Maldives in 2016.

## Discussion

We estimated the annual number of dengue cases arising among international air travelers from a set of 43 non-endemic countries to 119 dengue-endemic countries. Estimated infections were driven by a combination of traveler numbers and dengue risk, and the destination countries that were estimated to give rise to the most infections were India, Thailand, and the Philippines. The proportion of estimated symptomatic cases that were included in surveillance systems varied widely across countries suggesting that improvements in public and physician awareness as well as testing and reporting infrastructure may uncover more dengue cases among travelers.

These overall results can be interpreted primarily in two ways: First, assuming the predicted case numbers are accurate, varying reporting fractions reflects varying capacity of origin countries' surveillance systems to detect dengue cases among travelers. Second, and more likely, variation in reporting rates also reflects variation in infection hazard experienced by travelers relative to the endemic FOI that is not captured by the data (GDTM FOI and annual variation in dengue cases) or the model assumptions. For example, our analysis likely underestimated the number of cases among Australian travelers, and thereby overestimated the Australian reporting fraction. The presence of the USA among outliers with lower reporting fractions suggests that in certain years the model-predicted numbers of cases may be overestimated for this country. To understand this pattern, we looked at how reporting fractions varied by origin-destination country pairs and identified outliers for which case numbers could not be predicted well using data from other countries.

Travelers in outlying country pairs must experience different dengue risks or reporting probabilities than travelers from: i) other origin countries to the same destinations, or ii) the same origin country to other destinations. We found that

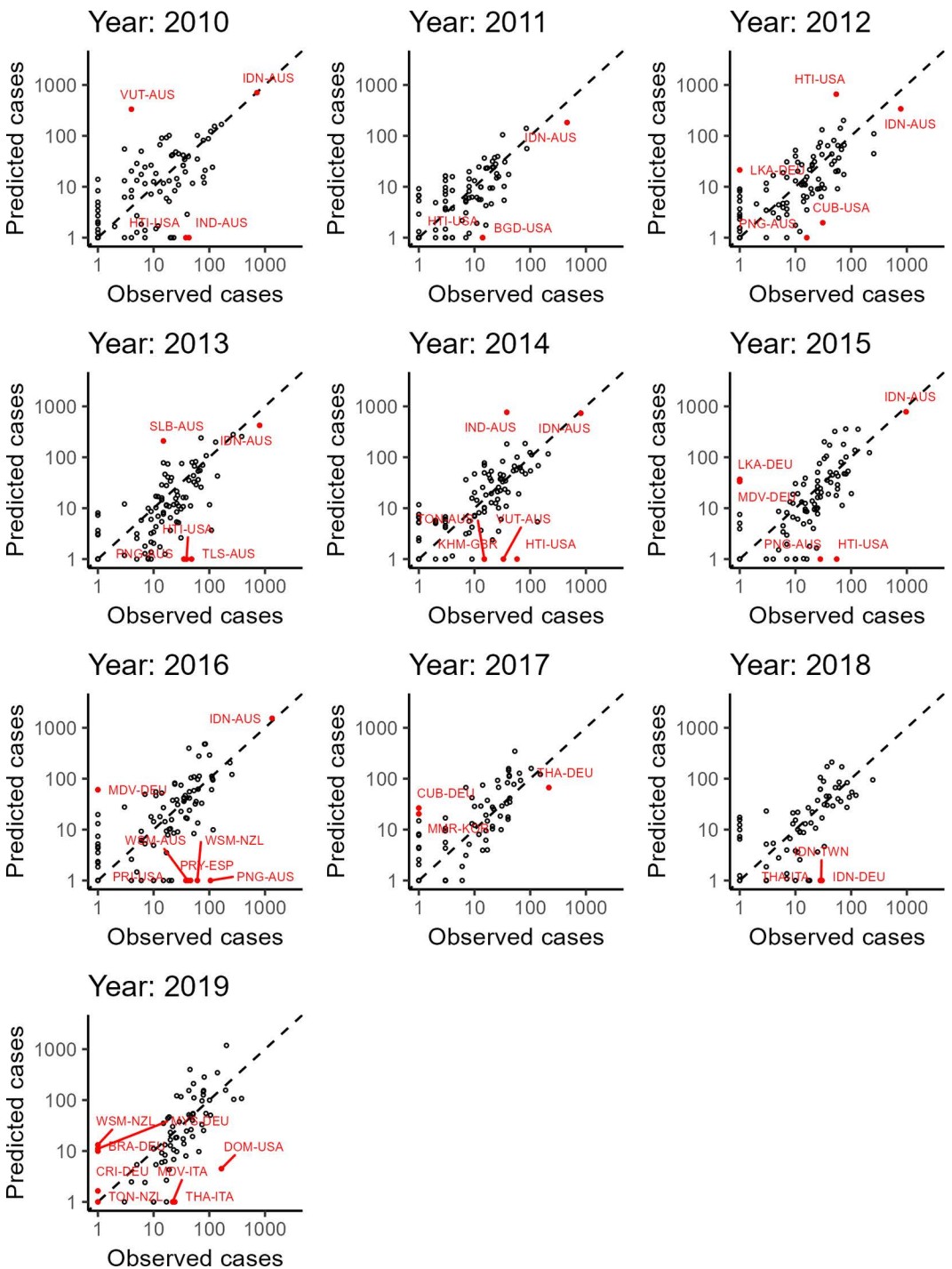

**Fig 3. Annual observed and predicted reported cases from leave-one-out validation.** The country pairs with the 100 highest total reported cases from 2010-2019 were left out one at a time, and reported cases by year were predicted from a model fit to the remaining data. Outliers/influential pairs are labeled in red, with the red line leading from the point itself to the label (Origin countries: AUS: Australia; DEU: Germany; ESP: Spain; GBR: United Kingdom/Great Britain; ITA: Italy; NZL: New Zealand; USA: United States of America. Destination countries: BGD: Bangladesh; BRA: Brazil; CRI: Costa Rica; CUB: Cuba; DOM: Dominican Republic; FJI: Fiji; HND: Honduras; HTI: Haiti; IDN: Indonesia; IND: India; JAM: Jamaica; KHM: Cambodia; LKA: Sri Lanka; MEX: Mexico; MDV: Maldives; MYS: Malaysia; PNG: Papua New Guinea; PRI: Puerto Rico; PRY: Paraguay; SGP: Singapore; SLB: Solomon Islands; SLV: El Salvador; THA: Thailand; TLS: Timor-Leste; TON: Tonga; VUT: Vanuatu; WSM: Samoa).

endemic island nations (Vanuatu, Solomon Islands, Papua New Guinea, Timor-Leste, Dominican Republic, Puerto Rico, Jamaica, Maldives, Haiti, Tonga, Fiji, and Samoa) were over-represented among outlier pairs, and thus did not align with the global average due to their highly variable annual dengue FOI which may not have been adequately captured by a combination of the GDTM FOI and publicly available dengue surveillance data. In these dengue endemic nations, FOI tends to have large year-to-year fluctuations [66,67] that are not adequately captured by surveillance systems, affecting our annual FOI estimates. Furthermore, when outbreaks do occur, as in Fiji in 2014 [68] or the Dominican Republic in 2019 [69], the number of predicted cases is lower than observed. Travelers from the USA make up 42% of reported cases acquired in island nations in this set of countries, and the Caribbean in particular represents the major source of travel-associated cases for US travelers [70,71]. Global models of dengue endemicity should be extended to adequately capture year-to-year variability in risk, which would be facilitated if age-specific data on dengue incidence were more widely available [67].

Several other studies have developed models of varying complexity to estimate dengue importation risk [30–41], but direct comparison is limited in several ways. Some studies did not use case data to validate or fit their models [32–34], while those that did predicted numbers of reported cases rather than the true burden of disease [32], probability of at least one importation [30], or total numbers of infections [36]. Of the studies that provided absolute numbers of cases, Quam et al. [33] estimated 572 travel-associated dengue cases annually in Rome from 2005-2012, compared to our estimate of 1,086 (95% CI 421–2,295) cases annually from 2010-2019 in Italy. Notably, Quam et al. based their risks of infection in destination countries on older work that indicated that the global incidence of dengue cases was twice as high as more recent estimates [72]. Our estimate of 7,298 (95% CI 2,800–15,665) annual travel-associated cases in China arising from 55 (out of 119) destination countries was substantially higher than Lai et al.'s [31] estimate of an average of 1,100 air travel-associated infections per year between 2005–2015 arising from nine countries in South-east Asia. In an *ad hoc* analysis, we estimated that an average of 5,630 (95% CI 2,192–12,339) travel-associated cases would arise from the same nine countries from 2010-2019, including 3,158 (95% CI 1,241–6,787) from 2010-2015. This is still substantially higher than Lai et al.'s estimate. Importantly, Lai et al. reported that the volume of airline travel into China was rapidly increasing between 2005–2015 (~4-fold increase), which could be a plausible reason for the higher estimates in our model for the period 2010–2015. Finally, La Ruche et al. [25] used a capture-recapture method to estimate that annual travel-associated dengue cases in France varied from 750-4,500 between 2007–2010. Our estimate of 3,040 cases (95% CI: 1,169-6,467) lies within this range.

Several findings were consistent across our study and the literature. Within Europe, Germany, Spain, Italy, France, and the UK were estimated to have relatively high numbers of imported dengue cases [32,36,37], while Southeast Asia and Brazil were important locations of infections for travelers [32,33]. Liebig et al. [36] estimated reporting probabilities of travel-associated dengue cases for Florida (1.4%), France (7.2%), Italy (9%), Spain (23.5%), and Queensland (28.6%) in 2015, which is the same ranking, and with estimates consistent with, our findings. Finally, Liebig et al. [36] found that Fiji and Taiwan were the two largest outliers for predicted imported cases to Queensland, similar to our finding that small island endemic locations were overrepresented among outliers.

Countries such as Mexico, Thailand and Indonesia are heterogeneous with respect to dengue FOI [73,74]. Therefore, the hazard of dengue infection from the GDTM may not be representative of the hazard of dengue infection experienced by an average traveler. To produce an FOI estimate per country, we aggregated across the country, weighting by population to represent the fact that travelers are more likely to go to population centers (i.e., a gravity-like model). However, travelers from certain countries may in fact go to higher-risk or lower-risk dengue areas. Australia and Germany, two countries with unusually high reporting rates possibly suggesting the model was underestimating cases, report a substantial fraction of cases from Indonesia relative to countries such as Spain and the USA. The island of Bali experiences some of the highest dengue rates within Indonesia [75] and it is also a popular tourist destination particularly for Australian travelers, which may explain the high reporting fraction for Australian and German travelers (49% and 44% respectively)

relative to other countries. Our approach aligns with most other dengue importation models in only considering a national average infection rate in each destination country [30–40]. Two studies considered subnational risks, where one did not account for the volume of travelers to each subnational region while the other did [41,76]. The latter study examined a special case where it was relatively straightforward to estimate the number of travelers to each subnational location, and the timing of their visits, because it only considered cases arising among foreign tourists attending the 2014 Brazil World Cup soccer matches [76]. However, future work could evaluate alternatives for assigning where travelers spend their time in destination countries (e.g., at Administrative Level 1 scales), especially for destinations that receive large numbers of travelers and exhibit substantial within-country heterogeneity in the risk of acquiring dengue (e.g., Thailand, Indonesia) by considering the airport used to arrive in the destination country.

A significant limitation of all models attempting to understand risk of dengue infection among travelers is the inability to distinguish between hazard and reporting probability as drivers of reported cases. It is likely that travelers could have different risks than residents, that these differences could be in either direction, and that they could differ based on the origin and destination countries of travelers. However, reported cases are a product of infection and reporting, so to distinguish between these mechanisms is challenging. With data on multiple countries, we were able to identify some countries that appear to contradict the assumption of constant hazard over time, and/or equal hazard to travelers and the average resident.

Multiple avenues of research could be pursued to gain a better understanding of the burden of dengue in travelers. With better spatial and temporal resolution of travel and reported case data, models such as this one could be used to disentangle the reasons that countries in this list have higher- or lower-than-average reporting fractions, such as Australia and Germany. Travel surveys could provide individual-level data on traveler behavior and knowledge and practices around dengue when traveling, while carefully designed, targeted seroepidemiological studies using sensitive assays could be used in conjunction with such surveys to validate the results of this model. Finally, in areas with limited but expanding local transmission such as Italy, France, and Florida, estimates from this model could be used to inform scenario models of local outbreaks initiated by travel-associated cases.

Our model makes simplifying assumptions which may introduce bias. We assumed that the duration of trips is the same for all travelers, whereas the number of infections arising from a distribution of durations would be smaller, and we imputed missing duration data. We also assumed that there was no seasonality in travel or in dengue FOI (as we had limited data on timing of travel-associated dengue cases with which to test this assumption), whereas if seasonality in travel is in or out of phase with dengue seasonality, this could lead to increases or decreases in infections [31,36]. Future models estimating global dengue FOI could include seasonality using trends in incidence reported by endemic countries' surveillance systems [60]. We also assumed that individuals are similarly susceptible to dengue infection and risk of symptoms, assuming that the proportion of travelers who are seropositive is small, whereas seropositive travelers are at lower risk of infection but a higher risk of symptoms if they become infected [77]. The much wider credible intervals for results where cases were the outcome rather than total infections indicates that future work would also benefit from more precise estimates of the proportion of infections that are symptomatic. An updated calculation of the proportion of infections that are symptomatic could change its point estimate too, with higher values increasing the burden calculated in Analysis 1 (case numbers) and lowering the results of Analysis 2 (reporting fractions). Finally, we only estimated numbers of cases among air travelers, while appreciable numbers of imported cases may occur in China and the USA among those who drive between neighboring countries.

This is the first study to assess the patterns in travel-associated dengue cases across such a wide range of dengue-endemic (n = 119) and non-endemic countries (n = 43). Our results can be useful during pre-travel health encounters to inform travelers and clinicians about the risks of dengue, where recommendations are often based primarily on reported case numbers [78,79]. A broader appreciation of variation in dengue risk across endemic countries might improve understanding of and adherence to recommendations based on case numbers. Increased detection of travel-associated cases may additionally mitigate the impact of local outbreaks in non-endemic countries with suitable mosquito vectors. Finally,

our research also highlights the need to improve surveillance systems in non-endemic countries. We extended previous models [35,38] by jointly estimating how reporting rates varied across dengue-endemic and non-endemic countries, thus highlighting the overall consistency of global trends in travel-associated cases and the utility of this model to perform prediction. We also found there is significant underreporting of symptomatic dengue cases among travelers, with a median of just 24.9% of such cases reported in travelers' origin countries. Although the per-trip risks of infection and illness are modest, the total burden of travel-associated dengue illness each year in the countries included in this study is substantial, and likely on the order of 60,000 cases and 300,000 total infections.

## Supporting information

**S1 Text. Supplementary Methods and Figures S1-S4.** Detailed methods and Supplementary Figures.
(DOCX)

**S1 Table. List of origin countries.**
(CSV)

**S2 Table. Publicly available reported dengue cases by origin country and year.**
(CSV)

**S3 Table. List of destination countries included.**
(CSV)

**S4 Table. Duration data obtained from UNWTO.**
(CSV)

**S5 Table. Estimated travel-associated dengue cases by origin country.**
(CSV)

**S6 Table. Estimated travel-associated dengue cases by destination country.**
(CSV)

**S7 Table. Estimated travel-associated dengue cases by origin country assuming constant dengue infection risk from 2010-2019.**
(CSV)

**S8 Table. Estimated travel-associated dengue cases by origin country from 2010-2015.**
(CSV)

**S9 Table. Estimated reporting fraction by origin country.**
(CSV)

**S10 Table. Estimated reporting fraction by origin-destination country pair.**
(CSV)

**S11 Table. Estimated reporting fraction by origin and destination country.**
(CSV)

## Author contributions

**Conceptualization:** Matt D. T. Hitchings, Justin J. O'Hagan, Derek A. T. Cummings.

**Data curation:** Matt D. T. Hitchings, Yi Xu, Bernardo García-Carreras, Adriana Gallagher.

**Formal analysis:** Matt D. T. Hitchings, Yi Xu, Bernardo García-Carreras.

**Investigation:** Matt D. T. Hitchings.

**Methodology:** Matt D. T. Hitchings, Derek A. T. Cummings.

**Project administration:** Justin J. O'Hagan.

**Resources:** Derek A. T. Cummings.

**Software:** Matt D. T. Hitchings, Yi Xu, Bernardo García-Carreras.

**Supervision:** Matt D. T. Hitchings, Derek A. T. Cummings.

**Visualization:** Matt D. T. Hitchings.

**Writing – original draft:** Matt D. T. Hitchings.

**Writing – review & editing:** Matt D. T. Hitchings, Yi Xu, Bernardo García-Carreras, Adriana Gallagher, Justin J. O'Hagan, Derek A. T. Cummings.

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
