## [Decision Letter · Decision Letter 0]

Response to Reviewers
Revised Manuscript with Track Changes
Manuscript

Shaden Kamhawi

co-Editor-in-Chief

Paul Brindley

co-Editor-in-Chief

**Journal Requirements:**

1) Please upload all main figures as separate Figure files in .tif or .eps format. For more information about how to convert and format your figure files please see our guidelines: 

2) Please amend your detailed Financial Disclosure statement. This is published with the article. It must therefore be completed in full sentences and contain the exact wording you wish to be published. 

**Reviewers' comments:**

**Key Review Criteria Required for Acceptance?**

**Methods**

-Are the objectives of the study clearly articulated with a clear testable hypothesis stated?

-Is the study design appropriate to address the stated objectives?

-Is the population clearly described and appropriate for the hypothesis being tested?

-Is the sample size sufficient to ensure adequate power to address the hypothesis being tested?

-Were correct statistical analysis used to support conclusions?

-Are there concerns about ethical or regulatory requirements being met?

Reviewer #1: See attached

Reviewer #2: -Are the objectives of the study clearly articulated with a clear testable hypothesis stated?

While the authors detail in their introduction the key results of their paper, the ultimate objective is not clearly presented. I recommend that the authors explain their goals clearly and how these results impact public health (1-2 sentences max).

-Is the study design appropriate to address the stated objectives?

Yes.

-Is the population clearly described and appropriate for the hypothesis being tested?

Yes.

-Is the sample size sufficient to ensure adequate power to address the hypothesis being tested?

Yes.

-Were correct statistical analysis used to support conclusions?

Yes.

-Are there concerns about ethical or regulatory requirements being met?

No ethical concerns.

**Results**

-Does the analysis presented match the analysis plan?

-Are the results clearly and completely presented?

-Are the figures (Tables, Images) of sufficient quality for clarity?

Reviewer #1: See attached

Reviewer #2: -Does the analysis presented match the analysis plan?

Yes.

-Are the results clearly and completely presented?

Yes.

-Are the figures (Tables, Images) of sufficient quality for clarity?

Yes. One smaller recommendation, see below.

**Conclusions**

-Are the conclusions supported by the data presented?

-Are the limitations of analysis clearly described?

-Do the authors discuss how these data can be helpful to advance our understanding of the topic under study?

-Is public health relevance addressed?

Reviewer #1: See attached

Reviewer #2: -Are the conclusions supported by the data presented?

Yes.

-Are the limitations of analysis clearly described?

Yes.

-Do the authors discuss how these data can be helpful to advance our understanding of the topic under study?

Yes, although I recommend that the authors add another short paragraph to their discussion highlighting in a structured manner future research directions, given their results.

-Is public health relevance addressed?

Yes.

**Editorial and Data Presentation Modifications?**

Reviewer #1: See attached

Reviewer #2: Only a minor non-mandatory suggestion, see point 8 of my general comments.

**Summary and General Comments**

Reviewer #1: See attached

Reviewer #2: Within this study, the authors seek to estimate dengue incidence in non-endemic countries via international air travel to endemic countries. This paper is interesting, topical and well-written. I recommend a minor revision be made to the following points before acceptance.

1. The ultimate purpose of this study is not clearly stated in the introduction. The Authors mention the main estimations provided by this study without mentioning their goals/implications.

2. From the discussion, it was not clear to me the future research directions foreseen by the Authors. The Authors do mention that "future model estimating... could use ..." as a hint to potential research avenues. However, I think this paper would greatly benefit from an additional paragraph that highlights what needs to be done in the future to build a framework that is directly beneficial to public health authorities (as indeed mentioned by the Authors in lines 394–397).

3. This paper's abstract does not present the authors' methodology clearly. There is a smaller reference to the data, but no mention of the methods being used. Please add a sentence to explain how you obtained your findings.

4. Author summary, line 54: add "considered" between "Across" and "countries" for clarity.

5. Author summary, line 56: "than previously understood". Please benchmark this previous understanding: what added value is this study bringing to the table?

6. Line 188, FOI's equation. Terms c_jt are undefined. Also, the force-of-infection is defined as "lambda" in the Supplementary Information. Please uniform your symbology.

7. Figure 2. I see error bars extending beyond 1. Is that physically plausible? If not, consider truncating the admissible interval to [0,1] and uniform results elsewhere too.

8. Figure 3. It is unclear how this model fit has been used. What's the meaning of red lines? Also, for the sake of clarity, have you considered one single scatter plot, where you put all these results together, and you plot each year's swarm in a different colour?

9. Discussion, line 292. Here you use an upper bound of 100%, with Figure 2 still extending beyond 100%. Please uniform.

10. Discussion, line 356. Please support your "low number of estimated cases" among Australian and German travellers with respect to other countries with a numeric figure.

11. Supplementary materials, mathematical model. The force of infection is here defined as lambda, different from the "FOI" symbol used in the main text. Also, please explain how these two definitions are related. Is lambda_k = FOI_k (and therefore computed upon terms c_jt)?

12. Supplementary materials, page 2, 2nd paragraph of Analysis 3. Here you use c^t_jk for the reported cases. But in Analysis 2, this quantity was defined as r^t_jk. Please uniform. Authors should be careful as symbol "c^t_jk" is similar to "c_jt" appearing in the calculation of the force of infection.

13. Model fit. It would be nice to have here a figure of their MCMC chains. Also, since the Author mention that they tested four different models, it would be useful to have a numerical explanation of why/how (the criteria used) they chose the "time-varying and no interaction" model against the others.

PLOS authors have the option to publish the peer review history of their article (what does this mean? ). If published, this will include your full peer review and any attached files.

**Do you want your identity to be public for this peer review?** For information about this choice, including consent withdrawal, please see our Privacy Policy .

Reviewer #1: No

Reviewer #2: No

**Figure resubmission:****Reproducibility:** To enhance the reproducibility of your results, we recommend that authors of applicable studies deposit laboratory protocols in protocols.io, where a protocol can be assigned its own identifier (DOI) such that it can be cited independently in the future. Additionally, PLOS ONE offers an option to publish peer-reviewed clinical study protocols. Read more information on sharing protocols at https://plos.org/protocols?utm_medium=editorial-email&utm_source=authorletters&utm_campaign=protocols

---

## [Decision Letter · Decision Letter 1]

Dear Dr Hitchings,

We are pleased to inform you that your manuscript 'Estimating the incidence of dengue in international air travelers from non-endemic countries between 2010-2019' has been provisionally accepted for publication in PLOS Neglected Tropical Diseases.

You will also note below that one of our reviewers suggested a careful reading and proofing of your manuscript for clarity. They pointed out one instance of potential confusion. This is particularly important since PLOS journals do not  offer copy edit services as part of the publication process.

Best regards,

Michael R Holbrook, PhD

Section Editor

Michael Holbrook

Section Editor

Shaden Kamhawi

co-Editor-in-Chief

Paul Brindley

co-Editor-in-Chief

Reviewer's Responses to Questions

**Key Review Criteria Required for Acceptance?**

**Methods**

-Are the objectives of the study clearly articulated with a clear testable hypothesis stated?

-Is the study design appropriate to address the stated objectives?

-Is the population clearly described and appropriate for the hypothesis being tested?

-Is the sample size sufficient to ensure adequate power to address the hypothesis being tested?

-Were correct statistical analysis used to support conclusions?

-Are there concerns about ethical or regulatory requirements being met?

Reviewer #1: No concerns

Reviewer #2: The manuscript meets the stated criteria for acceptance.

**Results**

-Does the analysis presented match the analysis plan?

-Are the results clearly and completely presented?

-Are the figures (Tables, Images) of sufficient quality for clarity?

Reviewer #1: No concerns

Reviewer #2: The manuscript meets the stated criteria for acceptance.

**Conclusions**

-Are the conclusions supported by the data presented?

-Are the limitations of analysis clearly described?

-Do the authors discuss how these data can be helpful to advance our understanding of the topic under study?

-Is public health relevance addressed?

Reviewer #1: No concerns

Reviewer #2: The manuscript meets the stated criteria for acceptance.

**Editorial and Data Presentation Modifications?**

Reviewer #1: I would just suggest the authors give the manuscript a careful read through. e.g. Line 28 ‘risks of infection while in 119 dengue-endemic locations (“destinations”),’ doesn’t make sense

Reviewer #2: None.

**Summary and General Comments**

Reviewer #1: (No Response)

Reviewer #2: I thank the authors for addressing my concerns, and I recommend this manuscript for acceptance.

PLOS authors have the option to publish the peer review history of their article (what does this mean? ). If published, this will include your full peer review and any attached files.

**Do you want your identity to be public for this peer review?** For information about this choice, including consent withdrawal, please see our Privacy Policy .

Reviewer #1: No

Reviewer #2: No